# Cognitive-behavioral statuses in depression and internet gaming disorder of adolescents: A transdiagnostic approach

**Rui She**[1], **Jiaxi Lin**[2], **Kei Man Wong**[2], **Xue Yang**[2]*

**1** Department of Rehabilitation Sciences, The Hong Kong Polytechnic University, Hong Kong, China,
**2** Center for Health Behaviours Research, JC School of Public Health and Primary Care, Faculty of Medicine, The Chinese University of Hong Kong, Hong Kong, China

☯ These authors contributed equally to this work.
* sherryxueyang@cuhk.edu.hk

**Data Availability Statement:** All relevant data are within the manuscript and its Supporting information files.

## Abstract

To investigate the comorbidity of adolescent depression and Internet gaming disorder (IGD) and their shared and unique cognitive-behavioral factors (i.e., self-esteem, dysfunctional attitudes, hopelessness, and coping), a large-scale school-based survey was conducted among 3147 Chinese secondary school students in Hong Kong. Probable depression and IGD were screened using the Centre for Epidemiological Studies-Depression Scale and DSM-5 IGD checklist, respectively. Multinomial logistic regression was performed to identify the associations between different condition statuses and cognitive-behavioral factors. Four groups were identified, including comorbidity group (having probable depression and IGD), IGD group (having probable IGD alone), depression group (probable depression alone), and healthy group (neither condition). Comorbidity group showed the worst cognitive-behavioral statuses, followed by depression group and then IGD group. Compared with healthy group, those with lower self-esteem and higher hopelessness and dysfunctional attitudes were more likely to be classified into depression group and comorbidity group, while maladaptive coping was positively associated with all three disorder groups. The results suggest that depression and IGD may share common cognitive-behavioral mechanisms (e.g., maladaptive coping) but also own their uniqueness regarding specific factors (e.g., hopelessness and self-esteem). A transdiagnostic intervention approach targeting the common factors may effectively address the comorbidity.

## Introduction

The prevalence of Internet gaming disorder (IGD), a common form of Internet addiction, is rapidly increasing, particularly among young individuals. IGD had been included as a condition warranting further research in *Section III of the Fifth Edition of the Diagnostic and Statistical Manual of Mental Disorders (DSM-5)* in 2013, including symptoms such as preoccupation, tolerance, withdrawal, and unsuccessful attempts to limit gaming [1]. IGD was found to affect 3.05% of the general population worldwide [2]. Compared with adults, adolescents are at

**Funding:** This study was funded by the Health and Medical Research Fund [#16171001 and #17180791] and General Research Fund [#14607319 and #14609820].

**Competing interests:** The authors have declared that no competing interests exist.

higher risk of IGD because of their immature cognitive control and may turn to Internet gaming as a maladaptive coping with stressful life issues and for escapism [3]. Asian adolescents often exhibit a higher prevalence of IGD than adolescents in Western countries [4] while the occurrence of IGD among Chinese adolescents varied between 2.4% to 21.5% [5–7].

## Comorbidity between IGD and depression

Behavioral addictions share many clinical manifestations and psychiatric comorbidities. Depressive symptoms were more common in individuals with IGD compared with the general population [8], and a reduction in depressive symptoms was observed in individuals remising from IGD [9]. Of all psychiatric comorbidities, depression showed the strongest correlation with problematic Internet use [10]. IGD and depression could be interrelated in different ways. Depressive symptoms may initiate or make permanent Internet addiction; Internet addiction may enhance psychiatric problems; Internet addiction and psychiatric symptoms may increase vulnerability to each other; and common risk factors can lead to both conditions [11]. Moreover, comorbid depression and addictive disorders lead to greater disease burden and poorer treatment outcomes compared with either condition alone [12]. Given the potential reciprocal mechanisms of comorbidity of depression and IGD, it is warranted to disentangle if some shared (or unique) modifiable factors of IGD and depression exist to inform health interventions in a transdiagnostic and cost-effective approach. Modifiable factors are those that can be controlled or altered through personal efforts and interventions. By preventing these factors, the risk of developing depression and IGD can be reduced. In contrast, non-modifiable factors such as age, sex, gene, or personality are less likely to be intervened effectively [13].

## Common factors of IGD and depression

The cognitive-behavioral theory is a widely used framework to understand depression [14] and addictive behaviors [15]. The theory indicates that cognition (think), emotion (feel), and behavior (act) are interconnected and collectively influence the onset and development of these disorders. Specifically, negative views of themselves (perceiving themselves as worthless, unlovable, inadequate, deficient), their environment (perceiving it as overwhelming, filled with obstacles and failure), and their future (perceiving it as hopeless, no effort will change the course of their lives) and maladaptive coping responses play central roles in the development and maintenance of these psychological and behavioral disorders, and symptoms can be reduced by changing such cognitive-behavioral statuses [16, 17]. Therefore, there may be overlapping cognitive-behavioral factors and shared mechanisms between depression and IGD. In this study, we investigated four plausible factors of depression and IGD, i.e., dysfunctional attitudes, self-esteem, hopelessness, and coping.

Dysfunctional attitudes are systems of beliefs that are unrealistic and perfectionistic in nature, which is regarded as a vulnerability factor for depression when life stress is heightened [18]. The primary construct domains of this depressogenic diathesis are perfectionism and the need for approval [19]. This pessimistic way of thinking shapes individual's perception, interpretation, and memory of personally relevant experiences, resulting in a negatively biased construal of one's personal world, and ultimately depressive symptoms. For instance, depression-prone individuals are more likely to notice and remember failed situations or ignore successful situations. Consequently, they maintain a negative sense of self (low self-esteem), leading to negative emotions and depression [20]. Dysfunctional attitudes and low self-esteem were found to predict the onset of depression during adolescence, which could persist into adulthood [21, 22]. Likewise, dysfunctional attitudes and low self-esteem were associated with a

higher risk of IGD, potentially using Internet gaming for escapism [23, 24]. A recent meta-analysis study identified that self-esteem had a significantly modest level of effect size on predicting IGD [13]. Thus, dysfunctional attitudes and self-esteem are plausible common factors of depression and IGD and contribute to the development of comorbidity of depression and IGD.

Hopelessness is another prominent cognitive factor of depression. Hopelessness defines the cognitive patterns characterized by pessimistic expectations regarding oneself and one's future and the helplessness about changing the situations [25]. The hopelessness theory posits that repeated exposure to uncontrollable and aversive environmental stimuli leads gradually to the belief that the aversive situation is inescapable and a sense of helplessness ensues regarding the situation. Hopelessness is a stressful status that was consistently demonstrated to result in depression [26] and be associated with relying on Internet gaming to escape from stressful real life [27].

Coping refers to cognitive and behavioral techniques used to manage internal or external demands, especially in stressful or challenging situations [28]. According to Lazarus's coping theory [28], evaluation and coping with stress are key to success in response to negative events or trauma. Adaptive coping strategies, which involve cognitive or active stress management, are believed to mitigate the negative effects of stress and safeguard against depression and risky behaviors in stressful situations [29, 30]. Maladaptive coping (in which the stressor is ignored or repressed), such as self-blame, catastrophizing, and rumination, is frequently used among people with depression and IGD [31–33].

## The present study

Although previous studies have tested the cognitive-behavioral factors of depression and IGD, respectively, few studies have explored their conjunctive effects on both disorders. Little is known about whether there are similar or different patterns of cognitive-behavioral statuses for depression and IGD; namely, whether the four groups divided based on the presence of depression and IGD [i.e., IGD alone, depression alone, IGD-depression (comorbidity group), and non-IGD-depression (healthy group)] would share the same or differentiated cognitive-behavioral mechanisms. The present study aimed to fill this research gap in a large-scale sample of Chinese adolescents. We hypothesized that there would be shared and unique cognitive-behavioral mechanisms underlying the two disorders.

## Materials and methods

### Participants

A school-based cross-sectional survey was implemented among secondary school students in Hong Kong (HK) from September to November 2020 using pencil-paper questionnaires when the schools were re-opened. As of the investigation time, all schools had been repeatedly required to suspend face-to-face school classes for almost five months for pandemic control. As reported previously, a stratified random sampling framework was used to select schools [31]; one secondary school was randomly chosen from each district of HK and invited to participate. Ultimately, 13 out of the invited 18 schools joined the study. All secondary 1–4 (7th-10th year of formal education) students were eligible if they a) were able to speak Chinese and b) had no learning disability. In light of the academic and exam pressures faced by Secondary 5 and 6 students, they were not included in the invitation for participation. The questionnaire took 15–20 minutes to complete. The questionnaire was completed and returned by 3147 out of 4323 participants, resulting in a response rate of 72.8%.

## Ethics

Both the participants and their parents provided consent for participation in the study. Participation is completely voluntary. The data was stored securely and anonymously which was accessible only to the research team. The research procedures were conducted following the principles outlined in the Declaration of Helsinki. The corresponding author's affiliation obtained ethics approval from the Survey and Behavioural Research Ethics Committee of the Chinese University of Hong Kong (#SBRE-18-433).

## Measurement

*Background variables*: The study collected sociodemographic information such as age, gender, place of birth (whether born in HK or not), living arrangement, and parents' educational level.

*Internet gaming disorder*: The Internet Gaming Disorder Short Form (IGD-SF) is a self-report scale consisting of nine items that assess IGD based on the criteria outlined in DSM-5. Response options include no (0) and yes (1) for each DSM-5 criterion. A higher score on the IGD-SF indicates a greater presence of symptoms related to IGD. If individuals fulfilled five or more conditions of these nine criteria within 12 months, they would be classified as suffering probable IGD [34]. The Chinese version of IGD-SF has been well-validated in adolescents [35]. The Cronbach's alpha was .77.

*Depressive symptoms*: The Centre for Epidemiological Studies-Depression (CESD) was used to assess participants' depressed affect, absence of positive affect or anhedonia, somatic symptoms, and interpersonal challenges during the last week. Higher total scores on the questionnaire indicate a higher level of depressive symptoms. As the meta-analysis suggested, the cut-off point of 20 was used to indicate the presence of probable depression [36]. The Cronbach's alpha of this scale was .85.

*Self-esteem*: The abbreviated version of the Rosenberg Self-esteem Scale (RSE) was used as the self-esteem measurement [37]. Studies suggested that RES was better specified as one positive dimension to reduce the overlapping of measures and response bias to negative-worded items [38, 39]. Therefore, this study selected the five positive items to evaluate self-esteem. The Cronbach's alpha of this scale was .90.

*Dysfunctional attitudes*: Six items from the Dysfunctional Attitude Scale Form A were used to measure the intensity of dysfunctional attitudes [40], which has shown good psychometric properties among Chinese people [41]. Sample item includes "If I don't set the highest standards for myself, I am likely to end up a second-rate person". Higher scores indicate higher levels of dysfunctional attitudes. The Cronbach's alpha of the scale was .87 in this study.

*Hopelessness*: The study utilized the 4-item Beck Hopelessness Scale (BHS) for measuring feelings of hopelessness, such as loss of motivation and expectations. This scale has been used among the Chinese population in HK and showed good psychometric properties [42]. Sample item includes "My future seems dark to me". High scores denote higher levels of hopelessness. The Cronbach's alpha was .51 (acceptable given the small number of indicators) [43].

*Maladaptive and adaptive coping*: The short version of the Cognitive Emotion Regulation Questionnaire (CERQ) was used to measure coping responses to stressful, threatening, or traumatic life events [44]. The CERQ consists of nine distinct subscales, such as self-blame, other-blame, rumination, and other related responses. Theoretically, the former four coping styles are deemed as maladaptive coping, while the last five categories are adaptive coping [44]. Adaptive and maladaptive coping subscale scores were obtained by summing up the scores belonging to the particular subscale. The higher the subscale score, the more a specific cognitive strategy was used. The Cronbach's alpha was .74.

## Statistical methods

The statistical analyses for the study were conducted using SPSS 26. Participants were firstly classified into four groups according to their depression and IGD status: healthy group (with neither probable depression nor probable IGD), IGD group (with probable IGD only), depression group (with probable depression only), and comorbidity group (with both probable depression and probable IGD). Descriptive data were presented as frequency, mean, and standard deviation (SD) stratified by the four groups. One-way analysis of variance was conducted to identify between-group differences in sociodemographic and cognitive-behavioral factors.

Multinomial logistic regression was used to identify the associations between sociodemographic factors and different condition statuses, using the group membership on the condition status as the dependent variable. The continuous cognitive-behavioral variables were firstly standardized as Z-factors which involves subtracting the mean and dividing by the standard deviation of each variable [45]. The associations between standardized cognitive-behavioral factors and probable depression/IGD status were investigated and adjusted for all sociodemographic factors. A set of comparisons among different groups was conducted. First, participants in the IGD group, depression group, and comorbidity group were compared with the healthy group to identify factors of different conditions. The second set of comparisons compared the IGD group and depression group with the comorbidity group. The final comparison was conducted between the IGD group and the depression group. Multivariate odds ratios (ORm) and the corresponding 95% confidence interval (CI) were presented. Additional analysis was also conducted to test the reliability and robustness of the findings. First, the analysis was repeated by using the cutoff point of 16 of CES-D to define depressive symptoms, which showed similar findings about the associations between cognitive-behavioral factors and conditions (S1-S4 Tables in S1 File & Fig 1). Second, generalized linear models (GLM) were used to test the potential effect of school clusters, which showed comparative findings with the multinominal logistic regression. P-values <0.05 were considered statistically significant.

## Results

### Characteristics of the sample

The participants had a mean age of 13.6 years old (SD = 1.3), with 48.1% being male. The majority of participants were born in HK (83.0%) and lived with both of their parents (73.0%). 15.7% of the participants' mothers and 13.5% of their fathers had obtained an educational level of college or above.

The mean CESD score was 19.9 (SD = 11.2) and the mean score was 2.3 (SD = 2.2) for IGD (Table 1). A total of 46.6% and 14.9% of participants reported probable depression and probable IGD, respectively. Of individuals with probable IGD, 69.5% had comorbid depression; 22.0% of the participants with probable depression had probable IGD. As shown in Fig 1, 10.3%, 4.5%, 36.3%, and 48.9% had both probable IGD and depression (comorbidity group), probable IGD alone (IGD group), probable depression alone (depression group), or neither condition (healthy group), respectively.

### Association between sociodemographic variables and different conditions

As shown in Table 2, compared with females, male adolescents were more likely to be classified in the IGD group (ORm = 2.27; 95%CI = 1.54–3.35; *p* <.001) and comorbidity group (ORm = 1.42; 95%CI = 1.10–1.84; *p* = .007) while less likely to be in the depression group (ORm = .51; 95%CI = .43-.60; *p* <.001). Adolescents who were not born in HK were more likely to be classified into the IGD group. In addition, living in a single-parent family and

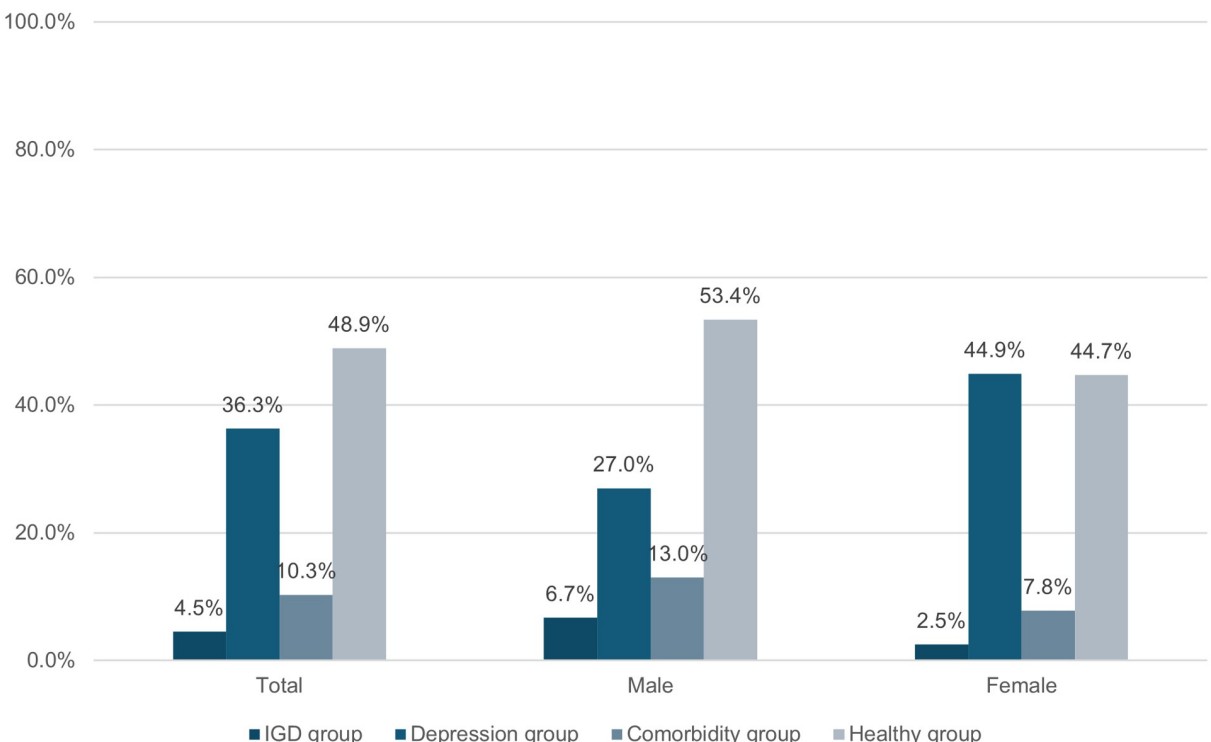

**Fig 1. Prevalence of IGD group, depression group, comorbidity group and healthy group by sex.** *Note: Total sample n = 3120 (27 cases with missing value on sex were excluded), Male n = 1498, Female n = 1622.

lower educational levels of fathers were associated with an increased likelihood of having probable depression alone or having comorbidity of probable depression and probable IGD. Educational levels of mothers did not show significant associations with the presence of probable depression or probable IGD.

## Associations between cognitive-behavioral variables and different conditions

As Table 3 shows, all the cognitive-behavioral variables, probable depression, and probable IGD were significantly correlated (range for r: .05-.60; all $p<0.01$) except for the correlation of adaptive coping with hopelessness and probable IGD. The results of multinomial logistic regression analyses in Table 4 showed that self-esteem was associated with a lower likelihood of classification in comorbidity group (ORm = .37; 95%CI = .31-.45) and depression group (ORm = .46; 95%CI = .40-.53), while hopelessness and dysfunctional attitudes were positively associated with comorbidity and depression group membership, controlling for sociodemographic factors. Maladaptive coping significantly differentiated between the healthy group and other groups as it was associated with an increased likelihood of having probable depression alone, probable IGD alone, and comorbidity.

Compared with participants with comorbidity, those in IGD group and depression group reported significantly higher self-esteem (ORm = 2.19; 95%CI = 1.66–2.89) and 1.25; 95% CI = 1.06–1.46), respectively) and lower dysfunctional attitudes (ORm = .44; 95%CI = .34-.59 and .68; 95%CI = .57- .81, respectively). Participants in IGD group also had lower level of

**Table 1. Comparison of socio–demographic, mental disorders, and cognitive–behavioral variables between different condition groups.**

| Independent variables | Total (N = 3147) | IGD group (N = 141) | Depression group (N = 1133) | Comorbidity group (N = 321) | Healthy group (N = 1525) | P-value |
|---|---|---|---|---|---|---|
| | N (%) | N (%) | N (%) | N (%) | N (%) | |
| *Sociodemographic variables* | | | | | | |
| Sex[a] | | | | | | <.001* |
| Male | 1507 (48.1%) | 100 (70.9%) | 404 (35.7%) | 194 (60.4%) | 800 (52.5%) | |
| Female | 1629 (51.9%) | 41 (29.1%) | 729 (64.3%) | 127 (39.6%) | 725 (47.5%) | |
| Born in Hong Kong[a] | | | | | | .013* |
| Yes | 2602 (83.0%) | 110 (77.5%) | 934 (82.4%) | 254 (78.9%) | 1290 (84.8%) | |
| No | 533 (17.0%) | 32 (22.5%) | 200 (17.6%) | 68 (21.1%) | 231 (15.2%) | |
| Live with parents[a] | | | | | | <.001* |
| Both | 2277 (73.0%) | 104 (73.2%) | 794 (70.3%) | 200 (62.9%) | 1168 (77.2%) | |
| Mother | 504 (16.2%) | 23 (16.2%) | 197 (17.4%) | 80 (25.2%) | 201 (13.3%) | |
| Father | 166 (5.3%) | 8 (5.6%) | 66 (5.8%) | 18 (5.7%) | 73 (4.8%) | |
| Neither | 171 (5.5%) | 7 (4.9%) | 72 (6.4%) | 20 (6.3%) | 71 (4.7%) | |
| Mother's educational level[a] | | | | | | .284 |
| Primary school or below | 226 (7.3%) | 15 (10.8%) | 86 (7.7%) | 28 (8.9%) | 97 (6.5%) | |
| Middle school | 1396 (45.3%) | 56 (40.3%) | 522 (46.6%) | 130 (41.4%) | 682 (45.7%) | |
| College or undergraduate | 408 (13.2%) | 21 (15.1%) | 133 (11.9%) | 39 (12.4%) | 213 (14.3%) | |
| Master or above | 78 (2.5%) | 6 (4.3%) | 29 (2.6%) | 10 (3.2%) | 33 (2.2%) | |
| NA (e.g., don't know) | 972 (31.6%) | 41 (29.5%) | 350 (31.3%) | 107 (34.1%) | 467 (31.3%) | |
| Father's educational level[a] | | | | | | <.001* |
| Primary school or below | 314 (10.2%) | 20 (14.3%) | 139 (12.4%) | 43 (13.5%) | 111 (7.4%) | |
| Middle school | 1449 (46.8%) | 57 (40.7%) | 529 (47.2%) | 150 (47.0%) | 708 (47.3%) | |
| College or undergraduate | 370 (12.0%) | 26 (18.6%) | 117 (10.4%) | 27 (8.5%) | 197 (13.2%) | |
| Master or above | 47 (1.5%) | 2 (1.4%) | 24 (2.1%) | 3 (0.9%) | 18 (1.2%) | |
| NA (e.g., don't know) | 913 (29.5%) | 35 (25.0%) | 312 (27.8%) | 96 (30.1%) | 464 (31.0%) | |
| | Mean (SD) | Mean (SD) | Mean (SD) | Mean (SD) | Mean (SD) | P-value |
| *Sociodemographic variables* | | | | | | |
| Age | 13.6 (1.3) | 13.5 (1.2) | 13.8 (1.3) | 13.6 (1.4) | 13.5 (1.3) | <.001* |
| *IGD symptoms* | 2.3 (2.2) | 6.2 (1.4) | 1.8 (1.4) | 6.3 (1.4) | 1.3 (1.3) | <.001* |
| *Depressive symptoms* | 19.9 (11.2) | 12.6 (4.8) | 29.1 (7.5) | 31.6 (8.5) | 11.2 (5.0) | <.001* |
| *Cognitive-behavioral variables* | | | | | | |
| Self-esteem | 14.2 (3.0) | 15.0 (2.6) | 12.9 (2.7) | 12.1 (3.2) | 15.5 (2.4) | <.001* |
| Dysfunctional Attitudes | 21.0 (7.3) | 19.5 (6.6) | 23.9 (6.5) | 26.4 (6.6) | 17.7 (6.3) | <.001* |
| Hopelessness | 13.1 (3.3) | 12.2 (2.7) | 14.7 (3.1) | 15.4 (3.2) | 11.5 (2.6) | <.001* |
| Coping | | | | | | |
| Adaptive | 16.3 (3.2) | 16.5 (3.2) | 16.5 (2.9) | 16.6 (3.4) | 16.1 (3.3) | .049* |
| Maladaptive | 12.9 (2.6) | 12.9 (2.5) | 13.6 (2.5) | 13.8 (2.6) | 12.2 (2.5) | <.001* |

*Note: one–way ANOVA test significant comparison (0.05)

[a]The number of individuals with missing values was 11 for sex, 12 for born in HK status, 29 for Live with parents' status, 67 for Mother's education level, 54 for Father's education level, for identifying different condition groups.

**Table 2. Associations between sociodemographic characteristics and statuses of depression and IGD using multinomial logistic regression.**

| Sociodemographic variable | Likelihood Relative to Healthy group | | | | | |
|---|---|---|---|---|---|---|
| | IGD group | | Depression group | | Comorbidity group | |
| | ORm (95% CI) | P-value | ORm (95% CI) | P-value | ORm (95% CI) | P-value |
| Sex | | | | | | |
| Female | 1.00 (reference) | | 1.00 (reference) | | 1.00 (reference) | |
| Male | 2.27 (1.54, 3.35) | <.001* | .51 (.43, .60) | <.001* | 1.42 (1.10, 1.84) | .007* |
| Born in Hong Kong | | | | | | |
| Yes | 1.00 (reference) | | 1.00 (reference) | | 1.00 (reference) | |
| No | 1.63 (1.06, 2.50) | .028* | 1.09 (.88, 1.36) | .425 | 1.32 (.96, 1.81) | .091 |
| Live with parents | | | | | | |
| Both | 1.00 (reference) | | 1.00 (reference) | | 1.00 (reference) | |
| Mother | 1.22 (.73, 2.02) | .451 | 1.38 (1.10, 1.74) | .006* | 2.11 (1.53, 2.90) | <.001* |
| Father | 1.33 (.61, 2.87) | .474 | 1.49 (1.04, 2.14) | .030* | 1.32 (.74, 2.33) | .348 |
| Neither | 1.32 (.59, 2.99) | .498 | 1.50 (1.05, 2.13) | .024* | 1.79 (1.06, 3.04) | .030* |
| Mother's education level | | | | | | |
| Primary school or below | 1.00 (reference) | | 1.00 (reference) | | 1.00 (reference) | |
| Middle school | .70 (.37, 1.36) | .295 | 1.04 (.75, 1.46) | .804 | .84 (.51, 1.37) | .482 |
| College or undergraduate | .75 (.34, 1.68) | .487 | .94 (.63, 1.41) | .757 | 1.010 (.55, 1.84) | .975 |
| Master or above | 1.50 (.46, 4.89) | .502 | 1.21 (.62, 2.36) | .581 | 2.16 (.85, 5.49) | .104 |
| NA (e.g., don't know) | .92 (.41, 2.07) | .840 | 1.19 (.80, 1.78) | .389 | 1.03 (.58, 1.85) | .913 |
| Father's education level | | | | | | |
| Primary school or below | 1.00 (reference) | | 1.00 (reference) | | 1.00 (reference) | |
| Middle school | .54 (.30, .98) | .041* | .58 (.43, .78) | <.001* | .59 (.38, .90) | .014* |
| College or undergraduate | .78 (.38, 1.63) | .515 | .501 (.34, .74) | <.001* | .34 (.18, .62) | <.001* |
| Master or above | .48 (.09, 2.68) | .401 | 1.02 (.48, 2.16) | .963 | .27 (.07, 1.11) | .069 |
| NA (e.g., don't know) | .40 (.19, .86) | .018* | .46 (.32, .67) | <.001* | .48 (.28, .81) | .006* |

Healthy group: no IGD and no depression.

ORm: multivariate odds ratio derived from the multinominal logistic regression model.

Odds ratios with p <.05 are presented with *.

hopelessness (ORm = .58; 95%CI = .44-.78) compared with comorbidity group. No significant differences in coping were observed between comorbidity group and depression/IGD group.

In addition, participants in IGD group reported higher levels of self-esteem (ORm = 1.76; 95%CI = 1.37–2.26) and lower levels of dysfunctional attitudes (ORm = .66; 95%CI = .52-.84)

**Table 3. Pearson's correlations between studied variables.**

| Variables | 1 | 2 | 3 | 4 | 5 | 6 | 7 |
|---|---|---|---|---|---|---|---|
| 1. Probable IGD | — | | | | | | |
| 2. Probable depression | .16* | — | | | | | |
| 3. Self-esteem | -.17* | -.43* | — | | | | |
| 4. Dysfunctional Attitudes | .19* | .44* | -.44* | — | | | |
| 5. Hopelessness | .17* | .47* | -.56* | .58* | — | | |
| 6. Adaptive coping | .03 | .06* | .16* | .05* | .02 | — | |
| 7. Maladaptive coping | .10* | .25* | -.10* | .32* | .26* | .60* | — |

*$p$<0.01.

**Table 4. Associations between cognitive–behavioral characteristics and statuses of depression and IGD using multinomial logistic regression.**

| Cognitive-behavioral variables | Likelihood Relative to Healthy group | | | | | | Likelihood Relative to Comorbidity group | | | | Likelihood Relative to Depression group | |
|---|---|---|---|---|---|---|---|---|---|---|---|---|
| | IGD group | | Depression group | | Comorbidity group | | IGD group | | Depression group | | IGD group | |
| | ORm (95% CI) | P-value | ORm (95% CI) | P-value | ORm (95% CI) | P-value | ORm (95% CI) | P-value | ORm (95%CI) | P-value | ORm (95% CI) | P-value |
| Self-esteem | .81 (.64, 1.03) | .087 | .46 (.40, .53) | <.001 | .37 (.31, .45) | <.001 | 2.19 (1.66, 2.89) | <.001 | 1.25 (1.06, 1.46) | .006 | 1.76 (1.37, 2.26) | <.001 |
| Dysfunctional attitudes | 1.19 (.95, 1.49) | .129 | 1.81 (1.59, 2.07) | <.001 | 2.68 (2.20, 3.27) | <.001 | .44 (.34, .59) | <.001 | .68 (.57, .81) | <.001 | .66 (.52, .84) | .001 |
| Hopelessness | 1.22 (.95, 1.56) | .129 | 2.13 (1.85, 2.44) | <.001 | 2.09 (1.72, 2.54) | <.001 | .58 (.44, .78) | <.001 | 1.02 (.86, 1.21) | .839 | .57 (.44, .74) | <.001 |
| Adaptive coping | 1.01 (.79, 1.30) | .931 | .98 (.86, 1.13) | .817 | 1.03 (.85, 1.24) | .779 | .98 (.74, 1.31) | .912 | .96 (.81, 1.13) | .614 | 1.03 (.79, 1.33) | .839 |
| Maladaptive coping | 1.38 (1.07, 1.77) | .014 | 1.56 (1.36, 1.80) | <.001 | 1.48 (1.21, 1.82) | <.001 | .92 (.69, 1.25) | .621 | 1.05 (.88, 1.26) | .569 | .88 (.68, 1.15) | .343 |

ORm: multivariate odds ratio derived from the multinominal logistic regression model adjusted for covariates.

Comorbidity group: both IGD and depressive symptoms; IGD group: IGD symptoms alone; Depression group: depressive symptom alone; Healthy group: neither condition.

and hopelessness (ORm = .57; 95%CI = .44-.74) compared with depression group. No differences in coping were observed between the two groups. Fig 2 further demonstrates the levels of cognitive-behavioral variables across the four condition groups.

## Subgroup analysis

We repeated the analysis of multinominal logistic regression between cognitive-behavioral factors and condition status, stratified by sex. The sex-specific analysis showed consistent results among males and females (S5, S6 Tables in S1 File).

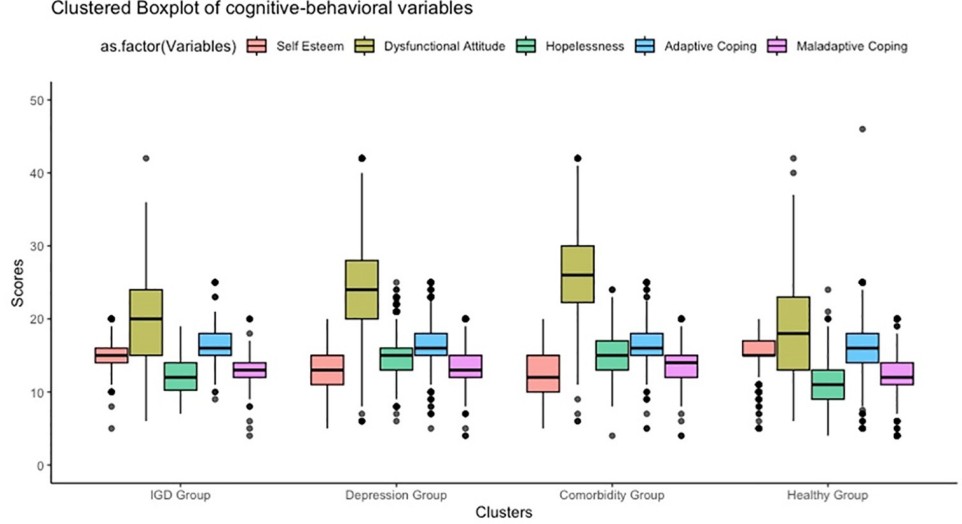

**Fig 2. Levels of cognitive–behavioral variables by clusters.**

## Discussion

The present study highlighted the high prevalence of probable depression and probable IGD in a large-scale sample of Chinese adolescents in HK, which was comparable to that reported in previous studies during the pandemic [46, 47]. The pandemic has caused significant disruptions to the daily life and educational routines of adolescents. Indeed, longitudinal studies indicate an increase in gaming use, IGD severity, and depressive symptoms among adolescents during the pandemic compared to pre-pandemic levels [48–50]. With school closures and social distancing measures, the Internet has become a main source of entertainment, communication, and learning for adolescents as well as a coping strategy for mental distress. In our sample, depression was a common comorbidity of IGD as 69.5% of adolescents with probable IGD had probable depression. This suggests that IGD and depression, as mental disorders, may share similar manifestations. However, only one-fifth of individuals with probable depression reported probable IGD, indicating that only a subset of adolescents with mental distress would turn to the virtual world and Internet gaming as a form of escapism. This is consistent with the extant literature, as individuals suffering from depression can exhibit a range of symptoms, including loss of passion, suicidal ideation, or other coping styles [51]. When treating adolescents with IGD symptoms, it could be beneficial to concurrently evaluate and manage any co-occurring mood symptoms (e.g., depression). This may enable healthcare providers to develop more holistic and effective intervention strategies. Practically, mental health professionals should notice that IGD is not always present as a singular diagnostic entity, but co-occurs with depression. Similarly, it is recommended to monitor the addictive behaviors of patients with mood disorders, such as Internet usage and substance use.

To better understand the developing mechanisms of IGD and depression and to facilitate the design of transdiagnostic and tailored interventions for the two disorders, we identified specific sociodemographic and cognitive-behavioral characteristics that might contribute to the comorbidity or differentiate individuals with different health statuses. Observed sex differences in the prevalence of probable depression and IGD suggest that females are more likely to suffer from internalizing/mood disorders and males tend to have externalizing/behavioral disorders [52], which may be explained by the differences in the biological (e.g., sex hormones) and sociocultural characteristics (e.g., gender roles, coping resources and tendency) [53]. Given these findings, the development of sex-specific interventions could potentially enhance the efficacy of treatments for these disorders. Consistent with previous studies, living with parents and high education levels of father were associated with a lower risk of comorbid depression and IGD [54, 55]. These findings suggest that better family environments and functions are likely to reduce the risk of adolescent psychological and behavioral problems, and family-based interventions (e.g., parent psychoeducation and family therapy) may be beneficial to reduce adolescents' IGD and depressive symptoms [56]. Such vulnerable groups (e.g., those living with single-parent families or having parents with low educational levels) should be paid particular attention from researchers and mental health professionals.

Furthermore, a combination of cognitive-behavioral characteristics was associated with conditions of depression and IGD. Compared with healthy group, maladaptive coping was independently associated with an increased likelihood of having probable depression alone, probable IGD alone, and comorbidity, while self-esteem, dysfunctional attitudes, and hopelessness were associated with having probable depression alone and comorbidity. A previous study suggested that there was one general psychopathology dimension that summarized individuals' propensity to develop any and all forms of common mental disorders [57]. Our findings provide evidence for maladaptive coping as a common attribute and mechanism underlying the development of persistent and comorbid mental health problems. Indeed,

these cognitive and behavioral responses towards stressors in daily life are vulnerabilities to both internalizing/mood and externalizing/behavioral disorders [58]. The results also provide empirical support for the application of the shared mechanism model of mental disorders [59]; specifically, it suggests that the cognitive-behavioral theory may be the shared mechanism for both depression and IGD [16, 17]. Maladaptive coping is modifiable and can be reduced effectively by CBT via coping skill training. CBT that addresses this common mechanism can be transdiagnostic to address a range of emotional disorders and addictive behaviors simultaneously [60]. Such an approach may be more cost-effective and efficient for disorders that have high comorbidity, such as IGD.

Furthermore, we identified some unique cognitive-behavioral characteristics of the three disorder groups. Compared with healthy group, depression group and comorbidity group, rather than IGD group, showed significantly lower self-esteem, higher hopelessness, and greater dysfunctional attitudes. It may suggest that the two groups shared more similar patterns in cognitive distortions. Self-esteem and distorted cognitions are related to the negative feelings of self and future, and thus may be more proximal to depressive symptoms, such as loss of interest and having self-harming thoughts, in the two groups [61]. The findings also imply that IGD symptomatology is more likely to be a manifestation of coping dysfunction instead of mood and emotional symptomatology. When using comorbidity group as the reference group, depression group and IGD group showed significantly better self-esteem and dysfunctional attitudes. The findings suggest that comorbidity group showed worse cognitive-behavioral statuses and should be paid particular attention by mental health professionals. Indeed, individuals suffering from multiple psychiatric conditions are more likely to have poor cognitive functions than those with a single psychiatric disorder [12]. Compared with depression group, IGD group showed better cognitive statuses, including lower hopelessness, higher self-esteem, and lower dysfunctional attitudes. It may imply that cognitive vulnerabilities are more relevant to mood disorders compared with addictive behaviors. It is also possible that Internet gaming might be a temporary escapism strategy from negative emotions. Maladaptive coping, however, was not able to differentiate comorbidity group from depression or IGD group, indicating that individuals in any of the three condition groups had a similar level of maladaptive coping. The findings may shed light on priorities when treating the three disorder groups. For example, tackling negative cognition may be more efficient to reduce depressive symptoms in depression group and comorbidity group than IGD group which do not show depressive symptoms, while reducing maladaptive coping styles can be equally prioritized for the three condition groups.

It is interesting to find that adaptive coping did not significantly differ in the four groups. It may imply that the protective effect of adaptive coping was not strong enough to reduce depression or IGD. Consistently, prior studies also reported non-significant roles of adaptive coping in mental health problems [62, 63]. It has been argued that maladaptive coping rather than adaptive coping might be more predictive of health problems [64]. More empirical studies are warranted to better understand the protective role of adaptive coping in both negative and positive health outcomes. It will help shed light on whether mental health interventions should enhance adaptive coping as a strategy.

## Clinical implications

The frequent co-occurrence of depression in adolescents with IGD suggests that the treatment of IGD and psychiatric comorbidities should be integrated into a cohesive service that minimizes both internalizing and externalizing symptoms. Untreated adolescent IGD and depression have been associated with various adverse outcomes that persist from adolescence into

adulthood, such as poor academic performance, social isolation, deteriorated physical health, and mental disorders in later life [65]. Hence, regular screening during adolescence is warranted for early detection and prevention. Moreover, the shared and unique cognitive-behavioral characteristics among groups with different condition statuses provide more nuanced information in the counseling and treatment for adolescents at risk of depression and IGD. Maladaptive coping may be a more universal and prioritized target for prevention and intervention of IGD and depression; whereas tackling negative cognitions (low self-esteem, hopelessness, and dysfunctional attitudes) may be more efficient to reduce depressive symptoms rather than IGD symptoms. Prevention and intervention should be tailored for adolescents with different profiles of conditions.

## Limitations

The present study has several limitations. First, the use of a cross-sectional study design does not allow for the establishment of causal relationships between variables. IGD, depression, and cognitive-behavioral characteristics may have reciprocal relationships. Longitudinal studies are greatly warranted to disentangle their causal relationships and predictors in preventing the development of single mental disorders or the comorbid depression and IGD. Secondly, short versions of dysfunctional attitudes and hopelessness scales were used to reduce the participants' response burden, and the hopelessness scale's internal reliability was relatively low. Therefore, future studies should validate the findings using full scales and more sophisticated measures for these variables. Thirdly, the data were derived from 1st to 4th-grade students in HK. Selection bias and generalization of findings should be cautious. However, sex distribution (male: 48.0% vs. 51.5%) and mean age (13.6 vs. 13.4 years old) of the current sample was comparable to the latest census data in HK [66]. Considering the significant emphasis placed on educational proficiency in HK, adolescents may face considerable academic pressures and high expectations from parents. This, coupled with the inherent stressors associated with residing in a densely populated city such as HK, may increase their vulnerability to depression and IGD [67]. Consequently, when extrapolating these findings to other regions, it is crucial to account for the contextual and geographical factors that may influence adolescent mental health in diverse areas. Fourth, this study used self-reported non-clinical diagnostic assessments for depression and IGD. False-positive cases might exist, and recall and social desirability bias could also affect the outcomes. Non-clinical samples were used and healthy group was defined as having neither probable depression nor probable IGD for the study purpose. Future work should validate the findings in clinical samples. Fifth, while the current study's scope is focused on IGD, other common Internet-related behaviors such as social media use were not assessed. However, it's important to note that social media use can have a significant impact on the mental health of adolescents and youth. It can enhance youth's social networks and serve as a source of social support [68]. Conversely, it may also induce detrimental psychological impacts such as depression and anxiety [50]. Hence, future studies should examine the interactions of different Internet behaviors and their impact on mental health. Lastly, we mainly focused on the modifiable intrapersonal factors based on the cognitive-behavioral theory. There may exist other important factors (e.g., personality and neurobiological, interpersonal, and environmental factors) and theories (e.g., social cognitive theory and mindfulness theory), which should be explored to understand adolescent IGD and depression and merit further research. The investigation of additional cognitive factors, such as mind wandering and cognitive flexibility [69], could also contribute to a more nuanced and comprehensive understanding of the mechanisms and determinants of adolescent IGD and depression. In addition, other comorbidities (e.g., anxiety, attention deficit hyperactivity disorder, substance

abuse, obsessive-compulsive disorder), positive health outcomes (e.g., psychological well-being), and their common and specific factors should be further investigated to better understand one's mental status as a whole and to facilitate a more efficient and comprehensive intervention for the patients.

## Conclusion

This study highlighted Chinese adolescents' vulnerability to both internal/emotional (i.e., depression) and external/behavioral health problems (i.e., IGD), and demonstrated their associations with various cognitive-behavioral characteristics. These findings supported the application of cognitive-behavioral theory to understand adolescent depression and IGD. The two disorders may share a common vulnerability of depression and IGD (i.e., high maladaptive coping) but also own their uniqueness regarding the specific factors (i.e., low self-esteem or hopelessness). Interventions targeting the identified common cognitive-behavioral variables may address adolescent depression and IGD transdiagnostically and cost-effectively.

## Supporting information

**S1 Data.**
(SAV)

**S1 File.**
(DOCX)

## Acknowledgments

We would like to thank Ms. Cindy Choi, Prof. Winnie Mak, and Prof. Joseph Lau for their assistance in data collection, and all the participants who dedicated their time to completing this study.

## Author Contributions

**Conceptualization:** Xue Yang.

**Data curation:** Kei Man Wong.

**Funding acquisition:** Xue Yang.

**Project administration:** Kei Man Wong.

**Supervision:** Xue Yang.

**Writing – original draft:** Rui She, Jiaxi Lin.

**Writing – review & editing:** Rui She, Jiaxi Lin, Xue Yang.

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
