## [Decision Letter · Decision Letter 0]

23 Feb 2024

PONE-D-23-37762Cognitive-behavioral statuses in depression and internet gaming disorder of adolescents: A transdiagnostic approachPLOS ONE

Dear Dr. Yang,

Thank you for submitting your manuscript to PLOS ONE. After careful consideration, we feel that it has merit but does not fully meet PLOS ONE’s publication criteria as it currently stands. Therefore, we invite you to submit a revised version of the manuscript that addresses the points raised during the review process.

We look forward to receiving your revised manuscript.

Kind regards,

Carmen Concerto

Academic Editor

PLOS ONE

Journal Requirements:

The Role of Nonverbal Cognitive Ability in the Association of Adverse Life Events With Dysfunctional Attitudes and Hopelessness in Adolescence - https://doi.org/10.1016/j.apnu.2012.02.004

In your revision ensure you cite all your sources (including your own works), and quote or rephrase any duplicated text outside the methods section. Further consideration is dependent on these concerns being addressed.

"This study was funded by the Health and Medical Research Fund [#16171001 and #17180791] and General Research Fund [#14607319 and #14609820]."

5. In this instance it seems there may be acceptable restrictions in place that prevent the public sharing of your minimal data. However, in line with our goal of ensuring long-term data availability to all interested researchers, PLOS’ Data Policy states that authors cannot be the sole named individuals responsible for ensuring data access (http://journals.plos.org/plosone/s/data-availability#loc-acceptable-data-sharing-methods).

7. Your ethics statement should only appear in the Methods section of your manuscript. If your ethics statement is written in any section besides the Methods, please move it to the Methods section and delete it from any other section. Please ensure that your ethics statement is included in your manuscript, as the ethics statement entered into the online submission form will not be published alongside your manuscript.

8. Please ensure that you refer to Figure 2 in your text as, if accepted, production will need this reference to link the reader to the figure.

9. We note you have included a table to which you do not refer in the text of your manuscript. Please ensure that you refer to Table 4 in your text; if accepted, production will need this reference to link the reader to the Table.

10. Please include captions for your Supporting Information files at the end of your manuscript, and update any in-text citations to match accordingly. Please see our Supporting Information guidelines for more information: http://journals.plos.org/plosone/s/supporting-information.

Reviewers' comments:

Reviewer's Responses to Questions

**Comments to the Author**

1. Is the manuscript technically sound, and do the data support the conclusions?

Reviewer #1: Yes

Reviewer #2: Yes

Reviewer #3: Yes

2. Has the statistical analysis been performed appropriately and rigorously? 

Reviewer #1: Yes

Reviewer #2: Yes

Reviewer #3: Yes

3. Have the authors made all data underlying the findings in their manuscript fully available?

Reviewer #1: No

Reviewer #2: No

Reviewer #3: Yes

4. Is the manuscript presented in an intelligible fashion and written in standard English?

Reviewer #1: Yes

Reviewer #2: Yes

Reviewer #3: Yes

5. Review Comments to the Author

Reviewer #1: This study, examining Internet gaming disorder, depression and psychological characteristics in a sample of Hong Kong adolescents, presents intriguing findings in a vital area of adolescent mental health research.

However, I have identified several areas where minor adjustments could significantly enhance the paper's rigor.

Measurement Tools and Internal Reliability of Hopelessness scale:

o Issue: The employment of abbreviated scales for measuring dysfunctional attitudes and hopelessness, alongside the lower internal reliability of the hopelessness scale, could potentially limit the depth and reliability of your findings.

o Suggested Action: I recommend explicitly addressing this in your limitations section, acknowledging the potential impact on your study's outcomes.

Geographical Focus:

o Issue: The study's concentration on a specific geographical area may restrict its broader applicability.

o Suggested Action: Detail this in the limitations section and elaborate on potential idiosyncrasies of your context in the implications section. This approach will provide a clearer understanding of how the findings might differ in other settings.

Data Sharing

Encouragement for Open Data:

o Issue: While respecting anonymity concerns, I suggest aligning more closely with the open data philosophy of this journal.

o Suggested Action: Consider creating an anonymized version of your dataset for public sharing. This will enhance the transparency and reproducibility of your research.

Extended discussion and bibliography enhancement

The discussion section of this study could be enhanced by addressing several key areas, while maintaining an updated and extensive comparison with prior studies:

a. Impact of the COVID-19 Pandemic: Given the timing of this study, it would be insightful to discuss the potential impact of the COVID-19 pandemic on adolescent mental health, particularly in relation to increased online activity and gaming during lockdowns. This discussion should consider how these factors might explain aspects of your data, especially concerning maladaptive coping strategies that could emerge in such situations.

i. Teng, Z., Pontes, H., Nie, Q., Griffiths, M., & Guo, C. (2021). Depression and anxiety symptoms associated with internet gaming disorder before and during the COVID-19 pandemic: A longitudinal study. Journal of Behavioral Addictions, 10, 169-180. https://doi.org/10.1556/2006.2021.00016.

ii. Sheen, A., Ro, G., Santos, A., Kagadkar, F., & Zeshan, M. (2020). 51.15 SCREEN TIME IN THE CONTEXT OF COVID-19: THE GOOD, THE BAD, AND THE UGLY. Journal of the American Academy of Child and Adolescent Psychiatry, 59, S255-S255. https://doi.org/10.1016/j.jaac.2020.08.425.

iii. Volpe U, Orsolini L, Salvi V, Albert U, Carmassi C, Carrà G, Cirulli F, Dell'Osso B, Luciano M, Menculini G, Nanni MG, Pompili M, Sani G, Sampogna G, Group W, Fiorillo A. (2022). COVID-19-Related Social Isolation Predispose to Problematic Internet and Online Video Gaming Use in Italy. International Journal of Environmental Research and Public Health, 19(3):1539. doi: 10.3390/ijerph19031539. PMID: 35162568; PMCID: PMC8835465.

iv. Marciano L, Ostroumova M, Schulz PJ, Camerini AL. (2022). Digital Media Use and Adolescents' Mental Health During the Covid-19 Pandemic: A Systematic Review and Meta-Analysis. Frontiers in Public Health, 9:793868. doi: 10.3389/fpubh.2021.793868. PMID: 35186872; PMCID: PMC8848548.

b. Gender Differences: A deeper exploration of gender differences in the experiences of depression and Internet Gaming Disorder (IGD) among male and female adolescents is warranted. This includes a separate analysis for each condition and its implications for targeted interventions, and how this is specifically manifested in your data.

i. Wang, R., Yang, S., Yan, Y., Tian, Y., & Wang, P. (2021). Internet Gaming Disorder in Early Adolescents: Gender and Depression Differences in a Latent Growth Model. Healthcare, 9. https://doi.org/10.3390/healthcare9091188.

ii. Phan, O., Prieur, C., Bonnaire, C., & Obradović, I. (2019). Internet Gaming Disorder: Exploring Its Impact on Satisfaction in Life in PELLEAS Adolescent Sample. International Journal of Environmental Research and Public Health, 17. https://doi.org/10.3390/ijerph17010003.

c. Long-term Consequences: The potential long-term consequences of untreated IGD and depression in adolescence should be discussed, including impacts on academic performance, social relationships, and future mental health, and how this relates to your findings.

i. Demirtas, O., Alnak, A., & Coskun, M. (2020). Lifetime depressive and current social anxiety are associated with problematic internet use in adolescents with ADHD: a cross-sectional study. Child and Adolescent Mental Health. https://doi.org/10.1111/camh.12440.

ii. Fumero, A., Marrero, R., Bethencourt, J., & Peñate, W. (2020). Risk factors of internet gaming disorder symptoms in Spanish adolescents. Computers in Human Behavior, 111, 106416. https://doi.org/10.1016/j.chb.2020.106416.

iii. Teng, Z., Pontes, H., Nie, Q., Xiang, G., Griffiths, M., & Guo, C. (2020). Internet gaming disorder and psychosocial well-being: A longitudinal study of older-aged adolescents and emerging adults. Addictive Behaviors, 110, 106530. https://doi.org/10.1016/j.addbeh.2020.106530.

d. Role of Social Media: Including the influence of social media on adolescent mental health, particularly in relation to self-esteem and depressive symptoms, would add value to the discussion. This should consider the intertwined nature of social media use and online gaming.

i. Fernandes, B., Biswas, U., Tan-Mansukhani, R., Vallejo, A., & Essau, C. (2020). The impact of COVID-19 lockdown on internet use and escapism in adolescents. Revista de Psicología Clínica con Niños y Adolescentes. https://doi.org/10.21134/RPCNA.2020.MON.2056.

ii. Nilsson, A., Rosendahl, I., & Jayaram-Lindström, N. (2022). Gaming and social media use among adolescents in the midst of the COVID-19 pandemic. Nordisk Alkohol- & Narkotikatidskrift: NAT, 39, 347-361. https://doi.org/10.1177/14550725221074997.

e. Mind Wandering: Discussing the role of mind wandering, a form of spontaneous thought not related to the task at hand, in adolescent mental health could be significant. This includes its relation to Internet Gaming Disorder (IGD), mental pain, self-reflection abilities, and other outcomes of interest.

i. Zhang, J., Zhou, H., Geng, F., Song, X., & Hu, Y. (2021). Internet Gaming Disorder Increases Mind-Wandering in Young Adults. Frontiers in Psychology, 11. https://doi.org/10.3389/fpsyg.2020.619072.

ii. Rodolico, A.; Cutrufelli, P.; Brondino, N.; Caponnetto, P.; Catania, G.; Concerto, C.; Fusar-Poli, L.; Mineo, L.; Sturiale, S.; Signorelli, M.S.; et al. (2023). Mental Pain Correlates with Mind Wandering, Self-Reflection, and Insight in Individuals with Psychotic Disorders: A Cross-Sectional Study. Brain Sciences, 13, 1557. https://doi.org/10.3390/brainsci13111557.

With these minor revisions to the discussion and associated bibliography, the article is in my opinion well-positioned for publication in this journal.

Reviewer #2: Thank you for the opportunity to review this work. Here, the authors investigated shared and unique cognitive-behavioral factors for depression and IGD among adolescents in Hong Kong secondary schools. It is an interesting read, and the topic is also timely.

The methodology employed aptly answers the research problem being investigated here.

Below are my specific comments that may be of help in strengthening the current paper:

1. Study procedures have been clearly described. Nevertheless, I would recommend the authors to describe the context of school disruptions caused by pandemic-related measures that coincided with the data collection period, and wherever applicable, also comment on their implications to the paper’s results.

2. The authors may consider to also report the 95% Cis when reporting odds ratios in the Results section.

3. p14, under the Discussion section: “When treating adolescents with IGD … other mood symptoms.”

This statement here does not seem to be well-supported by the results presented in this paper; the two disorders being comorbid may not imply anything about effectiveness of treatment. Clarification and/or justification on this point would be appreciated.

Minor comment:

- Fig. 2 – more horizontal separation between clusters may be helpful for visual interpretation.

Reviewer #3: This study found that depression and IGD may share common cognitive-behavioral mechanisms (e.g., maladaptive coping) but also own their uniqueness regarding specific factors (e.g., hopelessness and self-esteem). It is a novel transdiagnostic approach to explore the cognitive behavioral differences between comorbidity and solitary diseases. It provides a new idea for diagnosis and treatment.

However, there are some issues that need to be addressed as a transdiagnostic approach.

(1) Since IGD and depression have their own scales, and they are used as the standard to divide the categories in this study, then their cognitive behavioral differences were investigated. How does the cognitive behavioral scale increase diagnostic reliability and validity in this study? That is, the comorbidity can be diagnosed directly with IGD and depression scales without using cognitive behavioral scales.

(2) The particularity of comorbidity (compared with solitary disease, with healthy group) needs to be further highlighted to explain its mechanism and cause.

(3) Since comorbidity (321 people) suggested that IGA and depression share a common pathogenesis, why were there a large number of depressed people (1133 people) without IGA? Why were some IGAs (141 people) not depressed?

Further data analysis should be done to show which protective factors prevent a people solitary disease from developing into a comorbidity.

(4) Could IGA be a protective strategy for depression? It is suggested that this possibility be added to the discussion.

(5) This study only examined the cognitive behavioral scale and reached the mechanistic conclusion that cognitive behavioral variables were related to comorbidity. However, other scales were not examined, so other mechanisms cannot be ruled out. That is, if the other scale is used, will the mechanical conclusion of comorbidity and other variables be also obtained? The study did not test for exclusivity. It is suggested to discuss within limitations.

6. PLOS authors have the option to publish the peer review history of their article (what does this mean?). If published, this will include your full peer review and any attached files.

Reviewer #1: **Yes: **Pierfelice Cutrufelli

Reviewer #2: No

Reviewer #3: **Yes: **Jianxin Zhang

---

## [Author Response · Author response to Decision Letter 0]

18 Apr 2024

Dear Dr. Concerto,

Thank you for your email dated 24 Feb 2024 enclosing the reviewers’ comments. We sincerely thank the reviewers for their constructive and valuable comments, which were of great help in improving our manuscript. Accordingly, we have now systematically revised the manuscript by fully addressing and incorporating reviewers’ all comments. Our responses to the referees’ comments are given in a point-by-point manner below (we copy the reviewers’ comments first, followed by our responses). 

Response to Reviewer 1

Reviewer #1: This study, examining Internet gaming disorder, depression and psychological characteristics in a sample of Hong Kong adolescents, presents intriguing findings in a vital area of adolescent mental health research.

However, I have identified several areas where minor adjustments could significantly enhance the paper's rigor.

Measurement Tools and Internal Reliability of Hopelessness scale:

o Issue: The employment of abbreviated scales for measuring dysfunctional attitudes and hopelessness, alongside the lower internal reliability of the hopelessness scale, could potentially limit the depth and reliability of your findings.

o Suggested Action: I recommend explicitly addressing this in your limitations section, acknowledging the potential impact on your study's outcomes.

Response: Thanks for your suggestion. We have acknowledged it as a limitation.

“Secondly, short versions of dysfunctional attitudes and hopelessness scales were used to reduce the participants' response burden, and the hopelessness scale's internal reliability was relatively low. Therefore, future studies should validate the findings using full scales and more sophisticated measures for these variables.” (page 19)

Geographical Focus:

o Issue: The study's concentration on a specific geographical area may restrict its broader applicability.

o Suggested Action: Detail this in the limitations section and elaborate on potential idiosyncrasies of your context in the implications section. This approach will provide a clearer understanding of how the findings might differ in other settings.

Response: Thanks for your suggestion. We have added the discussion and acknowledged it as a limitation. 

“Thirdly, the data were derived from 1st to 4th-grade students in HK. Selection bias and generalization of findings should be cautious. However, sex distribution (male: 48.0% vs. 51.5%) and mean age (13.6 vs. 13.4 years old) of the current sample was comparable to the latest census data in HK [65]. Considering the significant emphasis placed on educational proficiency in HK, adolescents may face considerable academic pressures and high expectations from parents. This, coupled with the inherent stressors associated with residing in a densely populated city such as HK, may increase their vulnerability to depression and IGD [66]. Consequently, when extrapolating these findings to other regions, it is crucial to account for the contextual and geographical factors that may influence adolescent mental health in diverse areas.” (page 18)

Data Sharing

Encouragement for Open Data:

o Issue: While respecting anonymity concerns, I suggest aligning more closely with the open data philosophy of this journal.

o Suggested Action: Consider creating an anonymized version of your dataset for public sharing. This will enhance the transparency and reproducibility of your research.

Response: Thank you for the suggestion.We have uploaded the dataset as supplementary material for public access. 

Extended discussion and bibliography enhancement

The discussion section of this study could be enhanced by addressing several key areas, while maintaining an updated and extensive comparison with prior studies:

a. Impact of the COVID-19 Pandemic: Given the timing of this study, it would be insightful to discuss the potential impact of the COVID-19 pandemic on adolescent mental health, particularly in relation to increased online activity and gaming during lockdowns. This discussion should consider how these factors might explain aspects of your data, especially concerning maladaptive coping strategies that could emerge in such situations.

i. Teng, Z., Pontes, H., Nie, Q., Griffiths, M., & Guo, C. (2021). Depression and anxiety symptoms associated with internet gaming disorder before and during the COVID-19 pandemic: A longitudinal study. Journal of Behavioral Addictions, 10, 169-180. https://doi.org/10.1556/2006.2021.00016.

ii. Sheen, A., Ro, G., Santos, A., Kagadkar, F., & Zeshan, M. (2020). 51.15 SCREEN TIME IN THE CONTEXT OF COVID-19: THE GOOD, THE BAD, AND THE UGLY. Journal of the American Academy of Child and Adolescent Psychiatry, 59, S255-S255. https://doi.org/10.1016/j.jaac.2020.08.425.

iii. Volpe U, Orsolini L, Salvi V, Albert U, Carmassi C, Carrà G, Cirulli F, Dell'Osso B, Luciano M, Menculini G, Nanni MG, Pompili M, Sani G, Sampogna G, Group W, Fiorillo A. (2022). COVID-19-Related Social Isolation Predispose to Problematic Internet and Online Video Gaming Use in Italy. International Journal of Environmental Research and Public Health, 19(3):1539. doi: 10.3390/ijerph19031539. PMID: 35162568; PMCID: PMC8835465.

iv. Marciano L, Ostroumova M, Schulz PJ, Camerini AL. (2022). Digital Media Use and Adolescents' Mental Health During the Covid-19 Pandemic: A Systematic Review and Meta-Analysis. Frontiers in Public Health, 9:793868. doi: 10.3389/fpubh.2021.793868. PMID: 35186872; PMCID: PMC8848548.

Response: Thanks for your suggestion. We have extended the discussion on depression and IGD in the context of the pandemic and added some recommended references. 

“The present study highlighted the high prevalence of probable depression and probable IGD in a large-scale sample of Chinese adolescents in HK, which was comparable to that reported in previous studies during the pandemic [45, 46]. The pandemic has caused significant disruptions to the daily life and educational routines of adolescents. Indeed, longitudinal studies indicate an increase in gaming use, IGD severity, and depressive symptoms among adolescents during the pandemic compared to pre-pandemic levels [47-49]. With school closures and social distancing measures, the Internet has become a main source of entertainment, communication, and learning for adolescents as well as a coping strategy for mental distress.” (page 13)

b. Gender Differences: A deeper exploration of gender differences in the experiences of depression and Internet Gaming Disorder (IGD) among male and female adolescents is warranted. This includes a separate analysis for each condition and its implications for targeted interventions, and how this is specifically manifested in your data.

i. Wang, R., Yang, S., Yan, Y., Tian, Y., & Wang, P. (2021). Internet Gaming Disorder in Early Adolescents: Gender and Depression Differences in a Latent Growth Model. Healthcare, 9. https://doi.org/10.3390/healthcare9091188.

ii. Phan, O., Prieur, C., Bonnaire, C., & Obradović, I. (2019). Internet Gaming Disorder: Exploring Its Impact on Satisfaction in Life in PELLEAS Adolescent Sample. International Journal of Environmental Research and Public Health, 17. https://doi.org/10.3390/ijerph17010003.

Response: Thanks for your suggestion. We have compared the sex differences in the prevalence of depression and IGD in Table 1 and discussed the sex differences.

“Observed sex differences in the prevalence of probable depression and IGD suggest that females are more likely to suffer from internalizing/mood disorders and males tend to have externalizing/behavioral disorders [51], which may be explained by the differences in the biological (e.g., sex hormones) and sociocultural characteristics (e.g., gender roles, coping resources and tendency) [52]. Given these findings, the development of sex-specific interventions could potentially enhance the efficacy of treatments for these disorders. ” (page 14)

As this manuscript focused on the comorbidity of depression and IGD, we categorized the outcomes of depression and IGD into four groups. The results of the multinomial logistic regression analysis in Table 4 showed the associations between cognitive-behavioral factors and the likelihood of having individual or comorbid conditions. To further address the reviewer’s concern, we conduct an additional subgroup analysis by sex, examining the associations of cognitive-behavioral factors and status of depression and IGD as shown in Table 4. The findings for both female and male subgroups were relatively consistent and comparable to the results of the whole sample. Hence, for the simplicity of presentation, we mainly discussed findings derived from the total sample and uploaded the findings of sex-specific analysis as supplementary tables. 

c. Long-term Consequences: The potential long-term consequences of untreated IGD and depression in adolescence should be discussed, including impacts on academic performance, social relationships, and future mental health, and how this relates to your findings.

i. Demirtas, O., Alnak, A., & Coskun, M. (2020). Lifetime depressive and current social anxiety are associated with problematic internet use in adolescents with ADHD: a cross-sectional study. Child and Adolescent Mental Health. https://doi.org/10.1111/camh.12440.

ii. Fumero, A., Marrero, R., Bethencourt, J., & Peñate, W. (2020). Risk factors of internet gaming disorder symptoms in Spanish adolescents. Computers in Human Behavior, 111, 106416. https://doi.org/10.1016/j.chb.2020.106416.

iii. Teng, Z., Pontes, H., Nie, Q., Xiang, G., Griffiths, M., & Guo, C. (2020). Internet gaming disorder and psychosocial well-being: A longitudinal study of older-aged adolescents and emerging adults. Addictive Behaviors, 110, 106530. https://doi.org/10.1016/j.addbeh.2020.106530.

Response: Thanks for your suggestion. We have added the discussion on the long-term consequences of IGD when talking about clinical implications. 

“The frequent co-occurrence of depression in adolescents with IGD suggests that the treatment of IGD and psychiatric comorbidities should be integrated into a cohesive service that minimizes both internalizing and externalizing symptoms. Untreated adolescent IGD and depression have been associated with various adverse outcomes that persist from adolescence into adulthood, such as poor academic performance, social isolation, deteriorated physical health, and mental disorders in later life [64]. Hence, regular screening during adolescence is warranted for early detection and prevention.”(page 17)

d. Role of Social Media: Including the influence of social media on adolescent mental health, particularly in relation to self-esteem and depressive symptoms, would add value to the discussion. This should consider the intertwined nature of social media use and online gaming.

i. Fernandes, B., Biswas, U., Tan-Mansukhani, R., Vallejo, A., & Essau, C. (2020). The impact of COVID-19 lockdown on internet use and escapism in adolescents. Revista de Psicología Clínica con Niños y Adolescentes. https://doi.org/10.21134/RPCNA.2020.MON.2056.

ii. Nilsson, A., Rosendahl, I., & Jayaram-Lindström, N. (2022). Gaming and social media use among adolescents in the midst of the COVID-19 pandemic. Nordisk Alkohol- & Narkotikatidskrift: NAT, 39, 347-361. https://doi.org/10.1177/14550725221074997.

Response: We appreciate the reviewer’s suggestion. We have added the discussion on social media use in the limitation. 

“Fifth, while the current study’s scope is focused on IGD, other common Internet-related behaviors such as social media use were not assessed. However, it’s important to note that social media use can have a significant impact on the mental health of adolescents and youth. It can enhance youth’s social networks and serve as a source of social support [67]. Conversely, it may also induce detrimental psychological impacts such as depression and anxiety [49]. Hence, future studies should examine the interactions of different Internet behaviors and their impact on mental health.” (page 18-19)

e. Mind Wandering: Discussing the role of mind wandering, a form of spontaneous thought not related to the task at hand, in adolescent mental health could be significant. This includes its relation to Internet Gaming Disorder (IGD), mental pain, self-reflection abilities, and other outcomes of interest.

i. Zhang, J., Zhou, H., Geng, F., Song, X., & Hu, Y. (2021). Internet Gaming Disorder Increases Mind-Wandering in Young Adults. Frontiers in Psychology, 11. https://doi.org/10.3389/fpsyg.2020.619072.

ii. Rodolico, A.; Cutrufelli, P.; Brondino, N.; Caponnetto, P.; Catania, G.; Concerto, C.; Fusar-Poli, L.; Mineo, L.; Sturiale, S.; Signorelli, M.S.; et al. (2023). Mental Pain Correlates with Mind Wandering, Self-Reflection, and Insight in Individuals with Psychotic Disorders: A Cross-Sectional Study. Brain Sciences, 13, 1557. https://doi.org/10.3390/brainsci13111557.

Response: Thanks for your suggestion. We understand that mind wandering might be a relevant factor of adolescent mental health and IGD. However, it is not within the scope of the present study. We have added the discussion and acknowledged it as a limitation. 

 “Lastly, we mainly focused on the modifiable intrapersonal factors based on the cognitive-behavioral theory. There may exist other important factors (e.g., personality and neurobiological, interpersonal, and environmental factors) and theories (e.g., social cognitive theory and mindfulness theory), which should be explored to understand adolescent IGD and depression and merit further research. The investigation of additional cognitive factors, such as mind wandering and cognitive flexibility [68], could also contribute to a more nuanced and comprehensive understanding of the mechanisms and determinants of adolescent IGD and depression.” (page 19)

With these minor revisions to the discussion and associated bibliography, the article is in my opinion well-positioned for publication in this journal.

Response: We appreciate your suggestions for improving our manuscript. 

Response to Reviewer 2

Reviewer #2: Thank you for the opportunity to review this work. Here, the authors investigated shared and unique cognitive-behavioral factors for depression and IGD among adolescents in Hong Kong secondary schools. It is an interesting read, and the topic is also timely.

The methodology employed aptly answers the research problem being investigated here.

Below are my specific comments that may be of help in strengthening the current paper:

1. Study procedures have been clearly described. Nevertheless, I would recommend the authors to describe the context of school disruptions caused by pandemic-related measures that coincided with the data collection period, and wherever applicable, also comment on their implications to the paper’s results.

Response: Thanks for your suggestion. We have provided more contextual information on the school disruptions in HK during the pandemic in the Methods. The pandemic-related information is also available in our previous publication.

 “A school-based cross-sectional survey was implemented among secondary school students in Hong Kong (HK) from September to November 2020 using pencil-paper questionnaires when the schools were re-opened. As of the investigation time, all schools had been repeatedly required to suspend face-to-face school classes for almost five months for pandemic control. As reported previously, a stratified random sampling framework was used to select schools [30].” (page 7). 

Moreover, we have discussed the impact of pandemic on the interpretation of findings, 

“The present study highlighted the high prevalence of probable depression and probable IGD in a large-scale sample of Chinese adolescents in HK, which was comparable to that reported in previous studies during the pandemic [45, 46]. The pandemic has caused significant disruptions to the daily life and educational routines of adolescents. Indeed, longitudinal studies indicate an increase in gaming use, IGD severity, and depressive symptoms among adolescents during the pandemic compared to pre-pandemic levels [47-49]. With school closures and social distancing measures, the Internet has become a main source of entertainment, communicati

---

## [Editor Report · Decision Letter 1]

17 May 2024

Cognitive-behavioral statuses in depression and internet gaming disorder of adolescents: A transdiagnostic approach

PONE-D-23-37762R1

Dear Dr. Yang,

We’re pleased to inform you that your manuscript has been judged scientifically suitable for publication and will be formally accepted for publication once it meets all outstanding technical requirements.

Kind regards,

Carmen Concerto

Academic Editor

PLOS ONE
---

## [Editor Report · Acceptance letter]

23 May 2024

PONE-D-23-37762R1 

PLOS ONE

Dear Dr. Yang, 

I'm pleased to inform you that your manuscript has been deemed suitable for publication in PLOS ONE. Congratulations! Your manuscript is now being handed over to our production team.

Kind regards, 

on behalf of

Dr. Carmen Concerto 

Academic Editor

PLOS ONE